# The History of Live Attenuated *Centrin* Gene-Deleted *Leishmania* Vaccine Candidates

**DOI:** 10.3390/pathogens11040431

**Published:** 2022-04-02

**Authors:** Greta Volpedo, Parna Bhattacharya, Sreenivas Gannavaram, Thalia Pacheco-Fernandez, Timur Oljuskin, Ranadhir Dey, Abhay R. Satoskar, Hira L. Nakhasi

**Affiliations:** 1Departments of Pathology and Microbiology, Wexner Medical Center, The Ohio State University, Columbus, OH 43201, USA; volpedo.1@buckeyemail.osu.edu (G.V.); thalia.pachecofernandez@osumc.edu (T.P.-F.); 2Division of Emerging and Transfusion Transmitted Disease, Center for Biologics Evaluation and Research Food and Drug Administration, Silver Spring, MD 20993, USA; parna.bhattacharya@fda.hhs.gov (P.B.); sreenivas.gannavaram@fda.hhs.gov (S.G.); timur.oljuskin@usda.gov (T.O.); ranadhir.dey@fda.hhs.gov (R.D.)

**Keywords:** *Leishmania*, visceral leishmaniasis, vaccine, live attenuated parasite vaccines, pan-*Leishmania* vaccine

## Abstract

Leishmaniasis, caused by an infection of the *Leishmania* protozoa, is a neglected tropical disease and a major health problem in tropical and subtropical regions of the world, with approximately 350 million people worldwide at risk and 2 million new cases occurring annually. Current treatments for leishmaniasis are not highly efficacious and are associated with high costs, especially in low- and middle-income endemic countries, and high toxicity. Due to a surge in the incidence of leishmaniases worldwide, the development of new strategies such as a prophylactic vaccine has become a high priority. However, the ability of *Leishmania* to undermine immune recognition has limited our efforts to design safe and efficacious vaccines against leishmaniasis. Numerous antileishmanial vaccine preparations based on DNA, subunit, and heat-killed parasites with or without adjuvants have been tried in several animal models but very few have progressed beyond the experimental stage. However, it is known that people who recover from *Leishmania* infection can be protected lifelong against future infection, suggesting that a successful vaccine requires a controlled infection to develop immunologic memory and subsequent long-term immunity. Live attenuated *Leishmania* parasites that are non-pathogenic and provide a complete range of antigens similarly to their wild-type counterparts could evoke such memory and, thus, would be effective vaccine candidates. Our laboratory has developed several live attenuated *Leishmania* vaccines by targeted *centrin* gene disruptions either by homologous recombination or, more recently, by using genome editing technologies involving CRISPR-Cas9. In this review, we focused on the sequential history of *centrin* gene-deleted *Leishmania* vaccine development, along with the characterization of its safety and efficacy. Further, we discussed other major considerations regarding the transition of dermotropic live attenuated *centrin* gene-deleted parasites from the laboratory to human clinical trials.

## 1. Introduction

Leishmaniasis is the disease caused by the protozoan parasite *Leishmania* comprised of dozens of species of unicellular protozoan parasites. *Leishmania* parasites exhibit a diheteroxenous life cycle consisting of their bloodsucking sandfly insect host, which serves as a vector, and their primary vertebrate hosts [1]. The disease manifests in one of three main types that can be further divided into subtypes depending on the clinical manifestations. The three main types of leishmaniasis are cutaneous (CL), mucocutaneous (MCL), and visceral (VL), which are respectively characterized by lesions on the skin; destruction of the mucosa of the mouth, nose, and throat; or a potentially fatal infection of the visceral organs especially the spleen, liver, and bone marrow resulting in splenomegaly and hepatosplenomegaly. The type of pathology a host develops depends on the species of *Leishmania* and the interaction of that parasite with the host immune system [2,3,4]. Leishmaniasis is one of the top neglected tropical diseases (NTD) tracked by the World Health Organization (WHO) with an estimated 1 billion people at risk for infection and approximately 1.7 million infections occurring throughout 98 tropical and subtropical countries per annum [5,6]. An estimated 70,000 deaths per year also occur due to VL [2,3]. Leishmaniasis can be geographically divided into Old World and New World leishmaniasis, depending on the species causing the disease.

Treatment schemas for leishmaniasis consist mainly of various chemotherapies. Pentavalent antimonials (Sb^5+^) were the first chemotherapy drugs to be developed and used for the treatment of leishmaniasis [7]. However, after several decades of use, drug resistance to antimonial treatments has rendered them ineffective in many cases [2,7]. Several other drugs have been developed in response to antimonial drug resistance in *Leishmania* including Amphotericin B, pentamidine isethionate, miltefosine, and paromomycin [7]. The antimonial drugs and those developed as alternatives cause many side effects and do not always provide a clinical cure of leishmaniasis. At times, patients with severe side effects need to cease treatment until they recover and then slowly resume [2]. No vaccine is approved for use in humans to prevent leishmaniasis and many attempts to formulate a vaccine have failed in preclinical studies or did not demonstrate efficacy in phase I/II clinical trials [8]. Fortunately, cutaneous leishmaniasis can be treated with thermotherapy, but this is not feasible in the other forms of the disease [9]. These observations highlight the need of a successful *Leishmania* vaccine to alleviate the disease burden of leishmaniasis and avoid chemotherapies with high toxicity. Further, the World Health Organization and other regional authorities have set goals to achieve the elimination of leishmaniasis using various strategies. Therefore, the development of a safe and efficacious vaccine could help achieve the *Leishmania* elimination program. 

## 2. Vaccination Strategies against Leishmaniasis

An ancient practice of vaccination is leishmanization, in which intradermal inoculation of small amounts of virulent *Leishmania* (*L*.) *major* provides more than 90% protection against reinfection; this has been used in Israel, Iran, and the Soviet Union [10,11]. While leishmanization may potentially be an effective means of infection prophylaxis for some individuals, it is no longer practiced due to a range of safety and ethical reasons including non-healing skin lesions, exacerbation of skin diseases, the potential impact of immunosuppression, poor quality control, and the emergence of HIV, among others [12]. Scientific innovation and technological advances have paved the way for the development of efficient ways to generate an effective vaccine against leishmaniasis as an alternative to leishmanization. Although to this date there is no vaccine available against human leishmaniasis, a vaccine against different forms should, in theory, be possible [13]. Thus, over the years a multitude of vaccine strategies have been studied to control human leishmaniasis, which can be classified into first-generation vaccines (using dead parasites), second-generation vaccines (using genetically modified parasites expressing *Leishmania* genes encoding for recombinant proteins), and third-generation vaccines (plasmid-DNA based) [14]. For brevity, a discussion on all the experimental vaccines is not being attempted in this article.

### 2.1. First-Generation Vaccines

The first-generation vaccines composed of whole, killed parasites were used instead of leishmanization [15]. Several vaccination trials were conducted in many parts of the world using heat-killed whole *Leishmania* promastigotes as immunogens [16,17,18,19,20,21,22,23]. These vaccine candidates offer a huge repertoire of parasite antigens, and they can promote significant protection against infection [24,25,26,27]. Many studies exemplify the ability of a killed *Leishmania* vaccine adjuvanted with or without BCG to be a safe option [22,28,29,30,31,32]. However, this type of vaccine-induced immunity is not sustained and wanes over time, which resulted in reduced efficacy in clinical trials [15,26,29,30,33,34].

### 2.2. Second-Generation Vaccines

While the first-generation vaccines are still undergoing evaluation, additional research has also focused on second-generation vaccines. This includes trials using different recombinant molecules and parasite antigens including purified native protein fractions, bacteria or viruses carrying and expressing leishmanial genes, or even synthetic peptides representing epitopes of antigen [35,36]. These numerous subunits and recombinant *Leishmania* vaccine candidates utilize distinguished *Leishmania* antigens, including gp63, p36/LACK, A-2, FML, PSA-2/gp46/M-2, LCR1, ORFF, KMP11, LeIF, LmSTI1, TSA, HASPB1, protein Q, cysteine protease B(CPB), and A (CPA), and have been extensively reviewed [36,37,38,39,40]. Studies with many of these recombinant antigen vaccines in pre-clinical experimental models have shown the stimulation of a protective immune response against reinfection during CL and VL [38,40]. Notably, recombinant antigen vaccines such as LEISH-F1, LIESH-F2 (against CL), and LEISH-F3 formulated in GLA-SE adjuvant (against *L. donovani* and *L. infantum*) have reached Phase II clinical trials with promising results [41,42,43,44,45]. There has been much activity in the discovery of second-generation vaccine candidates for leishmaniasis since they pose no risk of infection and are suitable for immunocompromised individuals; however, difficulty in the isolation of enough purified protein for a subunit vaccine is a drawback that could hinder their widespread utilization [23,46]. *Leishmania* parasites are transmitted to a host during the sand fly bite, along with sand fly saliva. However, when these vaccines were tested against needle challenge versus sand fly transmission of a virulent parasite, they were either partially protective or not protective against the latter [47,48,49,50]. Further, pre-exposure to sand fly saliva has been shown to confer protection against vector-transmitted leishmaniasis, which raised the possibility to use sand fly salivary proteins as potential vaccine candidates either alone or combined with *Leishmania* proteins and is currently an expanding area of vaccinology research [51,52,53]. Nevertheless, a recent clinical study reported that immunity to vector saliva is compromised by short sand fly seasons in endemic regions of Tbilisi, Georgia, with temperate climates [54]. 

### 2.3. Third-Generation Vaccines

Third-generation vaccines utilize the direct injection of nucleic acids, comprised of either naked plasmid DNA or DNA encapsulated in a viral vector. Vaccinations with DNA encoding gp63, LACK, and PSA-2 all protected both genetically resistant and susceptible mice from infection with *L. major* [55,56,57,58,59], albeit they were not found to be protective against *L. chagasi* infection in BALB/C mice or against experimental VL [60,61]. Additional DNA vaccine candidates include genes encoding for proteins such as TSA, KMP11, A-2, NH36, LmSTI1, cysteine proteases, and histones have been shown to induce protective immunity in animal models and were previously reviewed [62]. In some studies, to improve the efficacy of DNA vaccines, the priming of these vaccines was accompanied by boosting the corresponding protein expressed on a recombinant viral vector such as vaccinia virus (VV) or modified vaccinia virus Ankara (MVA) [63,64,65,66]. A very recent Phase I clinical trial in 20 healthy volunteers reported that a Simian adenovirus (ChAd63) constituted of KMP-11 and HASPB, two genes of *L. donovani,* could effectively elicit a wide range of CD8+ T cells specific for *Leishmania* antigens and act as a promising approach for the prevention and treatment of human VL and post-kala-azar dermal leishmaniasis [67]. Importantly, the therapeutic effects of ChAd63-KH in Phase II of a non-randomized clinical trial are currently being evaluated [68]. Collectively, DNA vaccines in animal models are successful because they are stable, do not require adjuvants, produce the antigen over long periods, and can easily be produced in large quantities. [62]. However, concerns raised in relation to safety, such as the integration of the DNA into the mammalian genome and the induction of autoimmune diseases or cancer, have impeded the development of DNA-based vaccines for leishmaniasis [69]. 

### 2.4. Live Attenuated Vaccines

Live attenuated vaccines involve genetic modification of a parasite’s genes responsible for its virulence and/or survival and are of current interest in *Leishmania* vaccine research. It is an appealing approach because they closely mimic natural infection, ensuring the induction of immune responses consistent with protection without the danger associated with infection with live virulent parasites [70]. Early live attenuated *Leishmania* parasites developed through targeted deletion exhibited promise in inducing strong protection against homologous and heterologous challenge [71]. Because of the comparative ease of *Leishmania* whole-genome sequencing, along with the consequent capacity to monitor genome stability, there are fewer obstacles hindering testing genetically attenuated *Leishmania* parasites as candidate vaccines [72]. Numerous targeted gene deletions have been carried out recently to develop *Leishmania* live attenuated vaccine strains from *L. major*, *L. mexicana*, *L. amazonensis*, and *L. donovani*, which demonstrated significant protection against CL and VL in susceptible mice [73,74,75,76,77]. Notably, experimental mouse immunization with *Biopterin transporter 1* (*BT1*)-deleted *L. donovani* parasites [78] and *A2–rel* gene cluster in *L. donovani* [79] as well as *SIR2* [80], *Hsp70-II* [81], and *KHARON1* (KH1) *L. infantum* null mutants [82] as immunogens elicited protection against virulent challenge in BALB/c mice. Another *L. donovani* mutant attenuated for the p27 gene (*Ldp27^-/-^*), encoding an amastigote-specific cytochrome *c* oxidase component, resulted in reduced parasitic loads and long-lasting protection against CL and VL challenge [83,84]. Studies with several other defined attenuated *Leishmania* vaccines with specific gene targets such as *dhfr*, *lpg2*, *cpa*, *cpb*, *Ufm1*, and the paraflagellar rod-2 locus have shown differential protection in animal models [15,85,86]. For example, *dhfr-ts* null mutants of *L. major* could elicit significant resistance after BALB/c mice challenge with virulent *L. major* [87], while they were unable to induce protective immunity in primates [88]. *L. mexicana* mutants lacking cysteine proteinase genes resulted in delayed disease onset, smaller lesions, and lower parasite burden in mice and hamsters [89,90]. Similarly, phosphoglycan-deficient *L. major* immunization protected highly susceptible mice against virulent challenge without eliciting a strong Th1 response [91]. Summarizing numerous experimental evidence, Rivier et al. [92] concluded that injection of attenuated parasites achieved more heightened protection than any other method involving recombinant gp63 as a test antigen delivered with a variety of adjuvants and delivery systems. Collectively, it is interesting that similar gene-cloning technology that allowed the isolation of genes encoding vaccine candidates can now be used for engineering the ablation of essential genes in *Leishmania*, thus making the use of live attenuated vaccines a much more attractive proposition than ever before [93]. 

The use of live attenuated mutants as a prophylactic vaccine strategy is not an approach unique to leishmaniasis, but it has been successfully used for inducing protection against other parasitic diseases, such as malaria [94,95,96,97]. Live attenuated parasites lead to strong, long-lasting protection and could be utilized as a generalized strategy for all parasitic agents. 

Our laboratory has developed live attenuated *Leishmania* vaccines by targeted *centrin* gene disruptions, the sequential history of which will be elaborated in the following sections.

## 3. *Leishmania centrin* Knockout Parasites as a Vaccine Strategy against Leishmaniasis

### 3.1. Centrin-1-Deficient Leishmania donovani

*Leishmania centrin* is a calcium-binding protein located in the basal body of the parasite that regulates duplication or segregation of the centrosome. Specifically, *Leishmania* encodes five different *centrin* genes. Our group previously identified that one of them, the *Leishmania centrin* gene-1, is necessary for the growth and differentiation of the parasite, and its expression increases with differentiation from the promastigote to the amastigote stages [98]. Because of the role of *centrin* in sustaining parasite proliferation, it became an interesting target for deletion to generate a live attenuated *Leishmania* mutant with the potential to be used as a vaccine [98]. Hence, in 2004, our group developed a *centrin* null mutant of *Leishmania donovani* (*LdCen^-/-^*) by replacing the *centrin* gene with antibiotic-resistant genes. Microscopic analysis revealed a similar morphology between *LdCen^-/-^* and wild-type (WT) *L. donovani* promastigotes. Nevertheless, the story was much different in the amastigote stage, where centrin deficiency resulted in the lack of basal bodies’ formation and cytokinesis, which led to the induction of apoptosis and cell arrest at the G2/M phase. This inhibited amastigote growth also in in vivo infected macrophages [99]. It seems that, even when *LdCen^-/-^* can undergo kinetoplast and nuclear division, these parasites are incapable of complete cell division in the amastigote, forming large, multi-nucleated cells, resulting in apoptotic death of the parasite that can be easily cleared by the immune system [98,99]. Importantly, despite its major effects on amastigote replication, a lack of *centrin* does not affect promastigote replication in culture. Therefore, a live attenuated vaccine can be cultured even in high quantities, highlighting the relevance of these mutants in vaccine development [100]. 

In rodent studies, immunization with *LdCen^-/-^* generated long-lasting protection against the challenge with two WT *Leishmania* strains, *L. donovani* and *L. braziliensis*, causative agents of visceral and mucocutaneous leishmaniasis, respectively (Figure 1). Protection was demonstrated by the reduction in parasitic burden in the spleen and liver in comparison with non-vaccinated mice [101]. Moreover, *LdCen^-/-^* provides protection against *L. mexicana*, an American *Leishmania* strain causative of CL, observed by the lack of dermal lesion characteristic in non-immunized mice. Immunized mice showed a strong influx of MHC-II-expressing macrophages towards the *L. mexicana* challenge site, an increased number of IFN-γ-producing cells, and a reduction in IL-4 and IL-13 production, altogether characteristic of a protective inflammatory immune profile (Figure 1) [83]. It is important to highlight that *LdCen^-/-^* seems to provide a broad range of antigens that allow cross protection against different strains of *Leishmania*, including species endemic from different regions and causative of different manifestations of *Leishmaniasis*.

Moreover, *LdCen^-/-^* vaccine was demonstrated to be safe and effective even in immunosuppressed mice. This protection was correlated with an increase in both Th17 and Th1 immune responses [101,102]. CD4+ and CD8+ cells produced IL-1β, IL-6, TGF-β, and IL-23, cytokines necessary to sustain IL-17 production [102]. Additionally, there was an increase in Th1 cells, cytotoxic CD8+ T cells, and a higher ratio of IFN-γ/IL-10-producing T cells, as well as higher IgG levels, particularly of Th1-promoting IgG2a [101]. This is explained by the downregulation of the CD200-CD200R axis, an immune inhibitory pathway that is activated during WT *L. donovani* infection. However, infection with *LdCen^-/-^* limits the expression of CD200 in CD11c+ dendritic cells, resulting in the reduction in IL-10 and the increase in IFN-γ and TNF-α production by CD4+ T cells [103]. This potential live attenuated vaccine is effective even in aged mice, where protection against WT *L. donovani* was also provided, with similar, though less strong, immune response compared to young mice [104]. 

The adaptive immune response is favored and triggered by the innate immune response, particularly by macrophages, which are also the final host for *Leishmania* [105,106]. In order to favor its own survival, *Leishmania* modulates the immune response, usually by downregulating inflammatory pathways, leading to M1 polarization and, consequently, to Th1 activation [106,107]. Therefore, the polarization of macrophages after *LdCen^-/-^* vaccination is a determinant of protection. In vitro and in vivo models of infection showed that *LdCen^-/-^* parasites induce classical activation of macrophages towards an M1 phenotype. The M1 response was higher during *LdCen^-/-^* infection, compared to WT *L. donovani infection*, and was characterized by the upregulation of inflammatory mediator transcripts (IL-1β, IL-12, TNF-α, and iNOS2) and the downregulation of M2-related genes (*IL-10*, *YM1*, *Arg-1*, and *MRC-1* genes). Unlike WT parasites, *LdCen^-/-^* does not dampen the effector functions of macrophages and allows for the establishment of an M1 phenotype, which favors the anti-parasitic response by activating a protective adaptive immunity [108]. Additionally, it was recently reported by our group that *LdCen^-/-^* immunization results in a stronger recruitment of neutrophils to the infection site, mediated by CCL2 and CCL3, compared to its WT homologous strain [109]. This is not only important because neutrophils are the first responders against *Leishmania* infection [110] but also because the efficacy of *LdCen^-/-^* seems to be dependent on these cells. Particularly, we identified two distinct neutrophil subtypes, Nα and Nβ, where Nα could induce the ex vivo proliferation of T cells. This last population was observed in higher numbers after immunization with *LdCen^-/-^,* compared to infection with WT *L. donovani* [109]. Therefore, these results demonstrate that *LdCen^-/-^* induces strong innate immune responses, which are favorable for T cell anti-parasitic activity. 

### 3.2. L. donovani Cen^-/-^ for the Vaccination of Reservoirs of the Infection

Vaccination would not only prevent infected patients from developing severe forms of VL but can also protect infected asymptomatic individuals who do not receive drug treatment due to the lack of clinical evidence of infection. Asymptomatic individuals are considered to be reservoirs for the parasite and allow the continuous transmission of the disease. Therefore, vaccination would prevent new infections and, eventually, eradicate the disease [111]. Because asymptomatic infections are significantly more frequent than symptomatic cases, and asymptomatic individuals would likely be the predominant population receiving the vaccine, we considered that testing the live attenuated *LdCen^-/-^* vaccine in asymptomatic individuals would be of epidemiological relevance. Mice infected with WT *L. donovani* were used as a model for asymptomatic infection as they resembled the immunological conditions of asymptomatic individuals with high percentages of IFN-γ-producing cells and fewer IL-10-producing cells, as well as low parasitic burdens, characteristic of host protection. The central memory response was shown to be very similar after vaccination among asymptomatic mice and naïve mice, showing that *LdCen^-/-^* is immunogenic also in asymptomatic individuals. Moreover, these mutants generate protection against WT *L. donovani* challenge in asymptomatic mice, similarly to naïve mice vaccinated with *LdCen^-/-^*. Hence, this live attenuated vaccine could also be used to target asymptomatic individuals in endemic regions [112]. 

Months after VL clinical cure is diagnosed, some VL patients can develop a dermal manifestation of leishmaniasis called post-kala-azar dermal leishmaniasis (PKDL). PKDL occurs in 5–15% in the Indian subcontinent and up to 50% in Sudan of the patients apparently cured of VL [113]. Individuals with PKDL are considered an important reservoir as they can sustain anthroponotic transmission of *Leishmania*; therefore, targeting those individuals is considered necessary for the VL elimination program [114]. Therefore, *LdCen^-/-^* was tested in human cells of patients with healed VL and PKDL. After stimulation with *LdCen^-/-^*, macrophages obtained from PMBCs showed a strong production of IFN-γ, TNF-α, IL-2, IL-6, IL-12, and IL-17 and an increase in the percentage of IFN-γ-producing T cells (both CD4+ and CD8+) [115], hence demonstrating that *LdCen^-/-^* generates a protective Th1 immune response in humans, particularly in PKDL and healed individuals. 

Furthermore, the *LdCen^-/-^* vaccine has been tested in dogs, which are also an important reservoir for *L. infantum* in endemic regions. Dog vaccination is a necessary approach, as treating them generates drug resistance to the same drugs administered to humans. Currently, the recombinant vaccine Leishmune^®^ is commercially available for canine use. When *LdCen^-/-^* immunogenicity was compared to Leishmune^®^ in a canine model, a similar anti-*L. infantum* antibody response was observed, in both cases higher than in unvaccinated dogs. On the other hand, T cell responses were stronger in dogs vaccinated with *LdCen^-/-^,* specifically CD4+ and CD8+ T cell proliferation, CD8+ T cell activation, IFN-γ production, a general decrease in IL-4 production, and an increase in TNF-α and IL-12/IL-23p40 secretion by T cells. Therefore, the *LdCen^-/-^* vaccine is also immunogenic in dogs and it could also be used for preventing reservoir generation [116]. More importantly, *LdCen^-/-^* provided a long-term protection (24 months) against *L. infantum* in dogs and a reduction in parasitic burden in the bone marrow [117]. Although protection and antibody response were very similar to dogs vaccinated with Leishmune^®^, those vaccinated with *LdCen^-/-^* showed enhanced CD8+ T cell proliferation, a stronger Th1 response, and reduced IL-4 secretion [117,118]. Therefore, *LdCen^-/-^* could efficiently help control *L. infantum* transmission by dogs. 

### 3.3. L. major Cen^-/-^ as a Safer Alternative to L. donovani Cen^-/-^ and Leishmanization

Despite the high vaccine efficacy with *LdCen^-/-^*, several concerns remain, preventing the advancement of this mutant to human clinical trials. First of all, *LdCen^-/-^* parasites contain an antibiotic-resistance gene integrated in their genome [100]. In order to improve safety, a more efficient genetic engineering technique that does not require the insertion of antibiotic resistance, such as CRISPR-Cas9, had to be employed. Additionally, *L. donovani* WT is known to disseminate to the spleen and liver and cause visceral disease [119]. Due to its visceral nature and lack of visible clinical symptoms such as cutaneous lesions, VL often requires invasive procedures such as splenic and bone marrow aspirates in order to be diagnosed [120]. Once the disease is established, 95% of VL cases result in death when left untreated. Furthermore, the therapeutics available for VL often present toxicity and other side effects [119]. Because of these issues, the risks associated with the potential visceralization of *LdCen^-/-^* parasites prevent these mutants from being able to advance to clinical trials. Therefore, the use of a non-visceralizing, dermotropic *Leishmania* strain can improve safety. *L. major* is a particularly promising candidate as it is dermotropic and its infection presents with mild symptoms and a self-healing disease, compared to other cutaneous strains [121]. Furthermore, *L. major* has already been shown to effectively prevent subsequent infection through the practice of Leishmanization [11].

### 3.4. L. major Cen^-/-^ Mutants Provide Safe and Long-Term Immunity

Recently, our group developed a genetically engineered *L. major centrin* (*LmCen^-/-^*) gene knockout mutant strain using the CRISPR-Cas9 technique [8]. This technique allows for the targeted deletion of the *centrin* gene without the insertion of an antibiotic-resistance marker into the *Leishmania* genome, which is important for advancement to Phase I human clinical trials. These parasites are safe even in susceptible and immunocompromised mouse models, where *LmCen^-/-^* infection does not cause the development of lesions [8]. 

Immunization with *LmCen^-/-^* parasites elicited protection against intradermal needle challenge with virulent *L. major* [8]. Murine models physiologically relevant to human infection, such as the challenge with intradermal needle injection and sand fly challenge, are dependable approaches that simulate what would happen in endemic areas after vaccination. In particular, sand fly challenge mimics the natural route of infection and has been shown to modify the immune response to infection and vaccination. A previous study in particular showed that a killed *L. major* vaccine was able to protect against needle but not sand fly challenge; this was due to a population of neutrophils hindering vaccine efficacy [47]. Interestingly, *LmCen^-/-^* mutants also conferred protection against sand fly challenge [8], which has not been previously demonstrated in other *Leishmania* vaccine studies [47,48,49,50]. This protection was mediated by the rapid recruitment of multifunctional effector cells at the infection site, initiating a fast immune response after challenge [8]. Protection against sand fly challenge is a fundamental benefit for the advancement of these mutants to clinical trials as it is physiologically relevant to natural human infection.

Overall, immunization with *LmCen^-/-^* parasites was comparable to Leishmanization in conferring protection against needle or sand fly challenge with virulent *L. major* [8]. In particular, mice immunized with *LmCen^-/-^* parasites or healed from *L. major* WT (Leishmanization model) generated a comparable significantly higher pro-inflammatory immune response, characterized by IFN-γ+ effector T cells (Live CD4+CD44^Hi^Ly6C+T-bet+) and tissue resident memory (TRM) T cell response compared to the non-immunized challenged group (Figure 2) [8]. Taken all together, these results show that *LmCen^-/-^* mutants represent an equally effective but safer alternative to *L. donovani cen^-/-^* and Leishmanization for the protection against *L. major* WT needle and sand fly challenge.

Even though CL is the most common form of the disease, VL represents a much higher threat due to its potential to become fatal. Therefore, it is imperative that a vaccine against *Leishmania* is efficacious to prevent both cutaneous and visceral diseases. Recently, *LmCen^-/-^* mutants were shown to be safe in a hamster immunocompromised model, while in immunocompetent hamsters immunization was efficacious for the long-term protection against sand fly challenge with virulent *L. donovani* [122]. Immunized and challenged hamsters presented a pro-inflammatory profile characterized by significantly higher spleen expression of IFN-γ and TNF-α compared to non-immunized challenged hamsters (Figure 2) [122]. Furthermore, hamsters immunized with *LmCen^-/-^* parasites showed enhanced Th1 responses and decreased Th2 responses in the spleen compared to the group infected with *L. major* WT (Figure 2) [122]. Taken all together, these results revealed that *LmCen^-/-^* mutants are safe and effective vaccine candidates for pre-clinical models of CL and VL. However, the advancement of *LmCen^-/-^* mutants to clinical trials requires these parasites to be grown under current Good Manufacturing Practices (cGMP). Our group recently demonstrated that Good Laboratory Practices (GLP)-grade *LmCen^-/-^* parasites passed quality control and conferred long-term protection against *L. donovani* needle and sand fly challenges in hamsters [122]. These results were comparable to laboratory-grade *LmCen^-/-^* mutants. Furthermore, GLP-grade *LmCen^-/-^* parasites elicited the production of pro-inflammatory cytokines in human PBMCs [122], highlighting once more the potential of these parasites to serve as a candidate vaccine for human leishmaniasis.

### 3.5. Towards Developing a Pan-Leishmania Vaccine

Towards the goal of generating a pan-*Leishmania* vaccine, the efficacy of *LmCen^-/-^* parasites against challenge with other strains of *Leishmania* remains to be tested. Promising experimental data from *L. donovani* suggest cross protection among species. We hope to repeat these results using *LmCen^-/-^* mutants. Furthermore, along with *LmCen^-/-^* parasites, our group generated *L. mexicana* (a New World CL strain) *Cen^-/-^* mutants and tested them against homologous and heterologous challenges in mice [123] and hamsters (unpublished data). We found growth defect in *Cen^-/-^ L. mexicana* (*LmexCen^-/-^*) amastigotes analogous to what was previously observed with *L. major* and *L. donovani centrin*-deleted mutants. *LmexCen^-/-^ parasites* were also safe in immunocompromised mouse models and efficacious for the protection against *L. mexicana* wild-type parasite infection in genetically different BALB/c and C57BL/6 mice. Interestingly, while immunized BALB/c mice showed a decreased induction of Th2 responses, in C57BL/6 we observed increased Th1 responses in the skin and draining lymph nodes, suggesting a differential mechanism of protection depending on the host genetic background (Figure 2). Nevertheless, *LmexCen^-/-^* parasites showed long-lasting protection against homologous challenge with virulent *L. mexicana* by promoting pro-inflammatory immune responses and the generation of central T memory cells [123]. Immunization with the *LmexCen^-/-^* parasites also induced robust host protection in hamsters against *L. donovani* infection transmitted by sand fly bites (unpublished data). A *centrin* deletion mutant in *L. braziliensis* background was developed by another group recently using CRISPR/Cas9. Similar to *LdCen^-/-^*, *LmCen^-/-^*, and *LmexCen^-/-^* parasites, *LbCen^-/-^* mutants also exhibited a growth defect in the amastigote stage and did not cause cutaneous lesions in BALB/c mice. *LbCen^-/-^* parasites were undetectable in the ear, spleen, and lymph nodes by 6 weeks post-infection. Immunogenicity and protective efficacy characteristics of *LbCen^-/-^* parasites against virulent challenge remain to be determined [124]. Overall, *Cen^-/-^* parasites have shown promise as candidate vaccines against leishmaniasis; *LmCen^-/-^* mutants in particular are being advanced to human Phase I clinical trials.

## 4. Perspective

Several different vaccines against Leishmaniasis are currently being evaluated including newer RNA vaccines [125]. Beyond the demonstration of safety and efficacy characteristics in preclinical models, there is a new emphasis on identifying product development strategies towards the realization of a *Leishmania* vaccine. As some of the experimental vaccines enter large-scale production in the near future, cGMP manufacturing of *L. major* isolates in support of a Control Human Infection Model (CHIM) trial provided a framework to deploy manufacturing processes that are compliant with the expected regulatory standards [126]. In the absence of market incentives, the need for making a persuasive case in support of a *Leishmania* vaccine is being recognized. Due to the fluctuating nature of the case load of VL, Phase III trials to demonstrate the efficacy parameters may require creative strategies with respect to identifying and employing surrogate markers of efficacy since traditional measures of efficacy may be expensive and hard to realize. While CHIM studies make eminent sense for CL, currently there is no substantive equivalent for vaccine trials in VL as the parasites causing VL show a systemic dissemination and present a higher risk. However, CHIM studies have been used in *P. falciparum* malaria that also results in a systemic infection [127]. 

The efficacy observed in rodent models immunized with *LmCen^-/-^* that showed protection upon sandfly-mediated challenge with both *L. major* and *L. donovani* suggests that these mutants could be used as a pan-*Leishmania* vaccine [8,122]. Immunization with *LmCen^-/-^* has been shown to induce CD4+ T effector and tissue resident memory (TRM) T cell populations in frequencies comparable to leishmanization [8,128]. The data from leishmanization studies suggests that a certain level of parasitic persistence might be essential to maintain long-lasting protective immunity. Interestingly, BALB/c mice immunized with *LmCen^-/-^* and treated with dexamethasone showed some parasitic persistence at 15 weeks post-immunization [8]. A similar, low-level persistence was also found in STAT1 and STAT4 knockout mice immunized with *LmexCen^-/-^* [123]. Therefore, parasitic persistence at low frequency observed in our studies might be needed to mediate long-term protection, in addition to the memory cell populations [129]. These results suggest that immunization with *LmCen^-/-^* is a safer alternative to leishmanization with comparable protection and sharing a common immune mechanism of protection. A *Leishmania* skin test (LST) that measures the DTH response in a *Leishmania*-exposed individual may fulfill the need to measure the vaccine-induced immune response. LST was used in several vaccination studies but was discontinued due to the lack of a reliable supply and because it was never made under cGMP conditions [130]. The availability of a leishmanin reagent made under cGMP conditions may be a critical element in advancing *Leishmania* vaccine studies. Recent estimates indicate that 300–830 million doses for a vaccine preventing VL and 557–1400 million doses for a vaccine preventing CL over a 10-year period [131] would be needed. Such estimates are helpful in advancing the vaccine development efforts through agencies with a public health agenda in the endemic countries.

Additionally, the vaccine strategies described above for CL have not been explored against New World *Leishmania* species. For example, immunization with *LmCen^-/-^* has not yet been tested against the New World species of *Leishmania;* thus, its potential against these *Leishmania* species remains unrealized. To address that, we recently showed that a *centrin* deletion mutant in the *L. mexicana* background produced using CRISPR/Cas9-based targeting also conferred protection in pre-clinical animal models against homologous *L. mexicana* and against heterologous *L. donovani* challenge [123] (unpublished data). Taken together, using new technologies and exploiting established mechanism of leishmaniasis, it is possible to develop vaccine candidates that are safe and efficacious for human use against many forms of leishmaniasis. 

## Figures and Tables

**Figure 1 pathogens-11-00431-f001:**
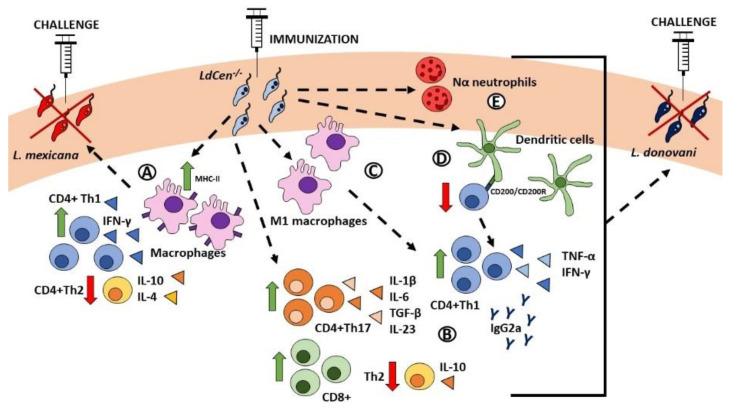
Immunity mediated by *centrin*-deficient live attenuated *L. donovani* mutants. (**A**) Immunization with *LdCen^-/-^* leads to a significant influx of MHC-II-expressing macrophages, resulting in higher levels of IFN-γ+-secreting CD4+ Th1 cells and lower levels of IL-10- and IL-4-sectreting CD4+ Th2 cells. This response provides protection against needle challenge with virulent *L. mexicana* parasites. (2–4) Immunization with *LdCen^-/-^* leads to a significant increase in IFN-γ- and TNF-α-secreting CD4+ Th1 cells, IL-1β-, IL-6-, TGF-β-, and IL-23-secreting CD4+ Th17 cells, CD8+ cytotoxic T cells, Ig2a, and lower levels of IL-10-secreting CD4+ Th2 cells (**B**). In particular, the increase in CD4+ Th1 cells is due to an accumulation of M1 macrophages (**C**) and the downregulation of the CD200-CD200R axis (**D**). These responses, along with the accumulation of pro-inflammatory Nα neutrophils (**E**) leads to protection against needle challenge with virulent *L. donovani* parasites. Abbreviations: MHC-II: major histocompatibility complex-II.

**Figure 2 pathogens-11-00431-f002:**
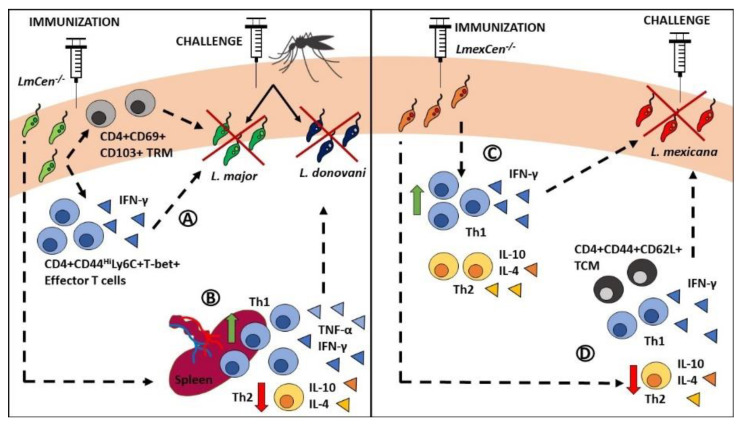
Immunity mediated by *centrin*-deficient live attenuated dermotropic *Leishmania* mutants. (**A**) Immunization with *LmCen^-/-^* leads to significantly higher levels of IFN-γ+ effector T cells (live CD4+CD44^Hi^Ly6C+T-bet+) and tissue resident memory T cells (CD4+CD69+CD103+) compared to non-immunized challenged mice. These cell populations mediate protection against needle and sand fly challenge with virulent *L. major* parasites. (**B**) *LmCen^-/-^* immunization also results in higher spleen expression of IFN-γ and TNF-α, enhanced Th1 responses, and decreased Th2 responses, which mediate protection against *L. donovani* challenge. (**C**) In C57BL/6 mice, immunization with *LmexCen^-/-^* leads to significantly higher Th1 responses in the skin and draining lymph nodes, compared to non-immunized controls, leading to protection against needle challenge with virulent *L. mexicana* parasites. (**D**) In BALB/c mice, immunization with *LmexCen^-/-^* leads to the generation of central memory T cells (CD4+CD44+CD62L+) and to significantly lower Th2 responses in the skin and draining lymph nodes compared to non-immunized controls, leading to protection against needle challenge with virulent *L. mexicana* parasites. Abbreviations: TRM: resident memory T cells; TCM: central memory T cells.

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
