# Peer review of "The History of Live Attenuated Centrin Gene-Deleted Leishmania Vaccine Candidates"

_pathogens, 2022, doi:10.3390/pathogens11040431_

Round 1

Reviewer 1 Report

In this Review, authors describe the history of use of live attenuated Leishmania vaccine candidates, focusing mainly in targeted deletion of centrin gene in this parasite. The proposal of this review is interesting and relevant, and just some minor points should be considered.

Minor points:

1) I would suggest the authors to include in their review a recent publication that used targeted deletion of centrin gene in Leishmania braziliensis, the most prevalent species in South America and responsible for cutaneous leishmaniasis. This is the first study that used this strategy in a Leishmania (Viannia) species. [Check at: https://pubmed.ncbi.nlm.nih.gov/35252020/]

2) Line 239: it is missing the verb in the sentence: “Also there an increase of…”

Author Response

Reviewer #1      

Comments and Suggestions for Authors

In this Review, authors describe the history of use of live attenuated Leishmania vaccine candidates, focusing mainly in targeted deletion of centrin gene in this parasite. The proposal of this review is interesting and relevant, and just some minor points should be considered.

Minor points:

1) I would suggest the authors to include in their review a recent publication that used targeted deletion of centrin gene in Leishmania braziliensis, the most prevalent species in South America and responsible for cutaneous leishmaniasis. This is the first study that used this strategy in a Leishmania (Viannia) species. [Check at: https://urldefense.com/v3/__https://pubmed.ncbi.nlm.nih.gov/35252020/__;!!KGKeukY!kZqaiE08n2D77DPzgNU0-pQQnAbzW6fYtc4ckLv-8r5-DUHcci4CioV41Mb1aLApKmIblC6u__4$ ]

Response: We thank the reviewer for suggesting this recent paper. We have added it in the manuscript in the section titled: “Towards developing a pan-Leishmania vaccine”. All changes are highlighted in yellow.

2) Line 239: it is missing the verb in the sentence: “Also there an increase of…”

Response: We apologize for the oversight. We have added the verb to this sentence.

Reviewer 2 Report

In this review, the authors summarize the studies carried out by Dr Nakhasi’s group in developing a vaccine against leishmaniasis based on genetically modified Leishmania lines lacking a centrin-coding gene. This group have made a huge characterization of this attenuate vaccine in different animal models; and in this manuscript, they recapitulate their findings. The manuscript is well-written and includes very informative schemes. The topic is really pertinent, as it is centred on a candidate vaccine against leishmaniasis that is close to entering clinical trials in humans. No vaccine against human leishmaniasis exist, but this is a very pursuit goal.

            The structure of the manuscript is fine. A brief description of the strategies followed for vaccine development against leishmaniasis is provided. Afterwards, the authors describe, following a historical perspective, the studies conducted by the group with this attenuated vaccine and the improvements introduced over time. Certainly, this is a notable contribution.

            There are two points that authors may consider to briefly comment:

  1. Persistence of the centrin gene deleted Leishmania parasites in the inoculated animals, and its possible relationship with the observed long-lasting immunity.
  2. A brief comment on the use of genetically attenuated parasites to prevent other parasitemias, e.g. malaria.

Author Response

Reviewer #2

Comments and Suggestions for Authors

In this review, the authors summarize the studies carried out by Dr Nakhasi’s group in developing a vaccine against leishmaniasis based on genetically modified Leishmania lines lacking a centrin-coding gene. This group have made a huge characterization of this attenuate vaccine in different animal models; and in this manuscript, they recapitulate their findings. The manuscript is well-written and includes very informative schemes. The topic is really pertinent, as it is centred on a candidate vaccine against leishmaniasis that is close to entering clinical trials in humans. No vaccine against human leishmaniasis exist, but this is a very pursuit goal.

            The structure of the manuscript is fine. A brief description of the strategies followed for vaccine development against leishmaniasis is provided. Afterwards, the authors describe, following a historical perspective, the studies conducted by the group with this attenuated vaccine and the improvements introduced over time. Certainly, this is a notable contribution.

            There are two points that authors may consider to briefly comment:

Persistence of the centrin gene deleted Leishmania parasites in the inoculated animals, and its possible relationship with the observed long-lasting immunity.

Response: We thank the reviewer for this comment. We have added a brief discussion of the importance of persistence of Cen-/- parasites in long-lasting protection in the “Perspective” section. The changes are highlighted in yellow.

A brief comment on the use of genetically attenuated parasites to prevent other parasitemias, e.g. malaria.

Response: We have added a few references about other live attenuated mutants used to prevent parasitic diseases in the “live attenuated vaccines” section. All changes are highlighted in yellow.